# Efficient Variational Continual Learning with optimal parameter trajectory in parameter hyperspace

## Abstract

Continual learning, a foundational challenge in machine learning, grapples with critical issues of efficient parameter storage and robust regularization, especially in Bayesian neural networks. Our research addresses these challenges with significant contributions. To address the storage complexity of fully connected layer parameters, we propose an efficient method that substantially reduces memory requirements. In convolutional neural networks, tailored parameter storage for Bayesian networks becomes essential, countering the parameter escalation caused by uncertainty inclusion. In variational continual learning, our work introduces an enhanced regularization term that preserves Kullback-Leibler divergence strengths while overcoming associated challenges. We also present an augmented Evidence Lower Bound term, crucial for capturing correlations between data and network parameters. This enables storage of common and distinctive parameter hyperspace bases, vital in continual learning. Our approach strategically divides the parameter subspace into common and distinctive subspaces, with conditions for effective backward and forward knowledge transfer, elucidating the network-parameter dataset correspondence. In summary, our contributions advance efficient and effective continual learning, offering insights into parameter storage and regularization techniques for Bayesian neural networks.

## 1 Introduction

Humans possess the cognitive ability to learn new concepts while retaining their previously acquired knowledge. Furthermore, the integration of novel knowledge can often result in an increased ability to understand and solve problems. Conversely, machine learning models have a limited capacity to retain previously learned problem-solving skills when trained to address new problems. In addition, these models lack the capability to enhance their problem-solving proficiency while learning to solve new tasks.

In order to tackle the issues of Catastrophic Forgetting Kirkpatrick et al. (2017), Li & Hoiem (2017) and promote Continual Learning, various approaches have been proposed in the literature. One such approach is the regularization-based model Kirkpatrick et al. (2017) Serra et al. (2018), which aims to preserve the previously learned knowledge by penalizing any modification of important parameters corresponding to previously learned tasks. Another approach is the dynamic architecture-based models Yoon et al. (2017), Rusu et al. (2016), Schwarz et al. (2018), which bifurcates different parameters of the models to different tasks. Additionally, memory-based models Wu et al. (2019), Lee et al. (2019), Castro et al. (2018) have been introduced that mitigate catastrophic forgetting by storing some raw samples Lopez-Paz & Ranzato (2017) of the previous tasks for rehearsal or using generative models to synthesize data from the past while training the current parameters. These approaches ensure that the knowledge of the past is maintained while learning new tasks.

After considering the above, a pertinent question arises: *can we enhance the problem-solving abilities of prior tasks using the knowledge of the problem-solving abilities of the current task?* In our approach, we do not include the data from past sessions for training the model, thereby eliminating the expansion-based and experience-based methods. Obtaining backward knowledge transfer in the traditional continual learning setting is nontrivial.

Point-based methods may have drawbacks, but Bayesian techniques (Blundell et al., 2015; Tseran et al., 2018) can avoid them by incorporating uncertainty in both parameters and network intermediates. Bayesian techniques handle uncertainty in parameters by representing each parameter as samples drawn from a distribution defined by its mean and variance. Among Bayesian techniques, Variational Inference (Hoffman et al., 2013) is the most famous, which approximates the prior and posterior distribution by minimizing the distance between the true and the approximate distribution using an ensemble of known distributions with the use of twice the number of parameters present in point-based regular neural networks. The introduction of additional uncertainty allows for flexibility in parameter learnability, as discussed in Ebrahimi et al. (2019). An approach to managing this uncertainty involves preserving highly uncertain parameters for ongoing learning, while keeping certain parameters fixed, as they have already acquired crucial attributes for the current task. However, this strategy introduces several drawbacks, particularly the challenge of controlling parameter learnability. Highly certain parameters may compel the updating of connected uncertain parameters during backpropagation as the parameters undergo updates. We propose utilizing the mean and variances of the parameter distribution to characterize the importance and uncertainty of parameters in the Continual Learning process with Bayesian neural network. Our approach involves carefully selecting the mean (position) and variance (spread) of the parameter distribution to accurately capture the uncertainty of the parameters. Subsequently, through meticulous experimentation, we visually illustrate the progression of the posterior distribution converging towards the optimal parameter distribution that is universally effective across all sessions. It can be intuitively understood that the more spread out (high variance) a parameter distribution is, the more commonality it shares with other session parameters, thus enabling the possibility of achieving commonality in the future to enhance parameter learnability.

Furthermore, within Bayesian techniques, optimization-based methods demonstrate superior performance compared to sampling-based approaches. Variational inference stands out as a prominent optimization-based method, enjoying widespread popularity in this category. Various variants of variational inference are present in the literature, among which Stochastic Variational Inference (SVI) (Hoffman et al., 2013) is noteworthy. In the domain of large datasets, SVI optimizes model parameters through the application of stochastic gradient descent. However, for effective backpropagation, a reliable gradient estimate of the evidence lower bound with respect to the variational parameters is essential. Black Box Variational Inference (BBVI) (Ranganath et al., 2014) emerges as a solution with minimal constraints on the form of the variational posterior, earning its designation as black box variational inference due to its flexibility. In our proposed method, we introduce an efficient approach by replacing the KL divergence term with regularization terms. This substitution facilitates gradient computation and imposes fewer constraints on the variational posterior within the context of variational continual learning.

**Contributions:** Our contributions can be summarized as follows. Firstly, we present an efficient method for storing fully connected layer parameters, significantly reducing storage complexity. Furthermore, we introduce an efficient parameter storage approach for convolutional neural networks. This is particularly beneficial in Bayesian networks where the parameter count increases due to the inclusion of parameter uncertainty. In the domain of variational continual learning, we propose an enhanced regularization term that preserves the positive attributes of KL divergence while effectively addressing associated challenges. For the fully connected neural network architecture, our focus is on minimizing the prior parameter size. This is achieved by integrating a compact network that adeptly learns the previous posterior weight distribution, demonstrating proficiency in generating samples from past distributions. Concerning convolutional neural networks, we leverage a representation matrix to optimize layer-specific parameters for each session. Furthermore, we introduce an augmented Evidence Lower Bound term, enabling the capture of correlations between data and network parameters. In our approach, we partition the parameter subspace into two subspaces. The common subspace demonstrates strong performance across all session datasets, while the other subspace contains distinctive information specific to each session. The augmented ELBO facilitates the storage of both common and distinctive parameter hyperspace bases, proving invaluable for continual learning tasks. Common bases contribute to overall performance across sessions, while distinctive bases offer novel insights into specific session characteristics. Additionally, we establish conditions that enable leveraging backward and forward knowledge transfer, while specifying the correspondence between network parameters and the dataset.

## 2 Related work:

**Continual Learning:** From a high-level view, Continual Learning models fall into memory-based, regularization-based, and gradient-based methods. Memory-augmented continual learning involves storing past data or features to enhance training, as seen in methods like Gradient Episodic Memory (GEM) (Lopez-Paz & Ranzato, 2017) and Averaged-GEM (A-GEM) (Chaudhry et al., 2018). Rehearsal-based models such as Experience-Replay (Rolnick et al., 2019) and Meta-Experience-Replay (Riemer et al., 2018) address forgetting by training on both previous and new data, requiring supplementary storage and adding to the overall cost. Regularization-based approaches penalize significant changes in important parameters from previous steps. Elastic Weight Consolidation (Kirkpatrick et al., 2017) and Synaptic Intelligence (Zenke et al., 2017) identify and assign importance scores to vital parameters. Gradient Projection Memory (Saha et al., 2021) and Orthogonal Gradient Descent (Farajtabar et al., 2020) leverage gradient steps aligned with important subspaces, stored in memory. Orthogonal Weight Modulation (OWM) (Zhang et al., 2021) modifies weights in directions orthogonal to past task inputs. Knowledge transfer is explored through Continual Learning with Backward Knowledge Transfer (CUBER) (Lin et al., 2022), allowing backward transfer if gradient steps align with previous task subspaces. Uncertainty-Guided Continual Learning (Ebrahimi et al., 2019) adjusts learning rates based on parameter uncertainty for improved backward transfer. In these methods, storing previous network parameters is essential, but none address efficient storage to reduce costs or establish universally beneficial knowledge across sessions.

**Bayesian approaches:** Research on integrating Bayesian techniques Mackay (1992), Snoek et al. (2012) into neural networks has been active for several decades. Several notable approaches in this field include Variational Inference with Gaussian Processes (Nguyen & Bonilla, 2014), Probabilistic Backpropagation (Hernández-Lobato & Adams, 2015), and Bayes by Backprop (Blundell et al., 2015). A method based on Variational Inference, known as Variational Continual Learning (Nguyen et al., 2017; Ahn et al., 2019), obtains the posterior distribution by multiplying the prior distribution with the data likelihood of the current session during training. Variational continual learning derives the current posterior by multiplying the previous posterior with the likelihood of the dataset. VCL incorporates supplementary coresets of the raw dataset, which are updated using the previous coreset and the current dataset. This approach updates the distribution with non-coreset datapoints, adding the challenge of managing coreset datapoints. Another variant proposed in Tseran et al. (2018) employs an online update of the mean and variance using Variational Online Gauss-Newton (Khan et al., 2018) update. Additionally, Chen et al. (2019) utilizes the Liu & Wang (2016) gradient to generate data samples for a known distribution within VCL.

## 3 Variational Inference framework:

In this section, we review the variational inference using a Bayesian neural network. In this framework, we use a Bayesian model $\mathbb{P}(\mathcal{Y}/\mathcal{X}, \mathcal{W})$ where we use a neural network model with weights $\mathcal{W}$ for the input-output combination of the dataset $\mathcal{D} = (\mathcal{X}, \mathcal{Y})$. We assume that the weight is drawn from a prior distribution $\mathcal{W} \sim \mathbb{P}(\mathcal{W}|\theta)$. During training obtaining the posterior using Baye's rule will lead to intractability. So Variational Inference obtains the approximate posterior with the following objective,

$$\mathcal{Q}(\mathcal{W}|\theta) = \arg\min_{\theta} \mathrm{KL}(\mathcal{Q}(\mathcal{W}|\theta)||\mathbb{P}(\mathcal{W}|\mathcal{D})) \tag{1}$$

or equivalently by minimizing the following loss function,

$$\mathcal{L}(\theta, \mathcal{Q}) = -\mathbb{E}_{\mathcal{Q}(\mathcal{W})}[\log \mathbb{P}(\mathcal{D})] + \mathrm{KL}(\mathcal{Q}(\mathcal{W}|\theta)||\mathbb{P}(\mathcal{W}|\mathcal{D})) \tag{2}$$

Here we take $\mathcal{Q}(\mathcal{W}|\theta)$ as Gaussian parameterized by $\theta = (\mu, \sigma)$.

### 3.1 Continual Learning with Variational Inference:

Consider a continual learning framework where data arrives sequentially, denoted as $\mathcal{D} = \{(\mathcal{X}_i, \mathcal{Y}_i)\}_{i=1}^{\mathcal{N}}$. Here, $\mathcal{X}_i = \{\mathcal{X}_{i,j}\}_{j=1}^{\mathcal{N}_i}$ represents input vectors, and $\mathcal{Y}_i = \{\mathcal{Y}_{i,j}\}_{j=1}^{\mathcal{N}_i}$ represents corresponding labels at session $i$. At time-step $t$, our objective is to learn the posterior $\mathbb{P}(\mathcal{W}_t/\theta, \tilde{\mathcal{D}}_t = \tilde{\mathcal{D}}_{t-1} \cup \mathcal{D}_t)$, where $\mathcal{D}_t = \{\mathcal{X}_{t,j}, \mathcal{Y}_{t,j}\}_{j=1}^{\mathcal{N}_t}$, and

$\mathbb{P}(\mathcal{W}_{t-1}/\theta, \tilde{\mathcal{D}}_{t-1})$ with $\tilde{\mathcal{D}}_t = \bigcup_{i=1}^{t} \mathcal{D}_i$. To update the parameter distribution at time step $t$, we lack access to $\tilde{\mathcal{D}}_{t-1}$. As discussed earlier, obtaining the exact posterior is intractable. Therefore, at each step, we assume $\mathcal{W}_t$ is sampled from an approximate posterior $\mathcal{Q}(\mathcal{W}_t/\theta_t) \approx \mathbb{P}(\mathcal{W}_t/\alpha, \tilde{\mathcal{D}}_t)$. Our objective is to minimize the following loss function:

$$\mathcal{L}_{\text{ELBO}}(\theta_t, \mathcal{D}_t) = \mathbb{E}_{\mathcal{Q}(\mathcal{W}/\theta_t)}[-\log \mathbb{P}(\mathcal{D}_t/\mathcal{W}_t)] + \text{KL}(\mathcal{Q}(\mathcal{W}_t/\theta_t)||(\mathcal{Q}(\mathcal{W}_{t-1}/\theta_{t-1})) \qquad (3)$$

Here, Our model weight $\mathcal{W}_i$ gets updated at each time-step such that $\mathcal{W}_t \sim \mathcal{Q}(\mathcal{W}_t|\theta_t)$.

## 4 Efficiently learning prior weights in parameter space

The existing Bayesian methodology for computing the likelihood term in Equation 3 relies on Monte-Carlo sampling. However, this approach is highly sensitive to the sample size, governed by the law of large numbers, and a small sample size can lead to inaccurate estimates. This dependence on sample size not only compromises accuracy but also escalates both space and time complexity, posing challenges to computational efficiency. Furthermore, the network, in this Bayesian framework, is compelled to store the prior distribution $\mathcal{Q}(\mathcal{W}_{t-1}/\theta_{t-1})$ using $\theta_{t-1} = (\mu_{t-1}, \sigma_{t-1})$. Consequently, the number of parameters to be maintained becomes precisely twice the size of the network weights. We leverage a Bayesian neural network wherein the parameter $\theta$ governing $\mathcal{Q}(\mathcal{W}|\theta)$, where $\theta = (\mu, \sigma)$, is learned through stochastic variational inference (Hoffman et al., 2013) and back-propagation. This approach allows for the derivation of a more tightly bounded lower limit on the variational objective, showcasing accelerated performance in contrast to other sampling-based Bayesian techniques, such as Markov Chain Monte Carlo Blei et al. (2017). To comprehensively assess the effectiveness of our approach we carefully examine two models: a fully connected neural network and a convolutional neural network. These models are known for their effectiveness in computer vision tasks, and we're exploring how they perform differently in Bayesian learning because of their distinct architectures.

### 4.1 Efficient Parameter Update for Fully Connected Neural Networks:

Consider a fully connected neural network operating in a supervised learning scenario, where each input and corresponding label training data pair originates from a distinct session in the training dataset, denoted as $\mathcal{D}$. Let $\mathcal{X} \in \mathbb{R}^n$ represent the input vector, $\mathcal{Y} \in \mathbb{R}^m$ denote the label vector in the dataset. Specifically, $\theta = \{\mu, \sigma\}$ consists of $\{\mu, \sigma\} = (\{\mathcal{W}_\mu, \mathcal{B}_\mu\}, \{\mathcal{W}_\sigma, \mathcal{B}_\sigma\})$, where $(\mathcal{W}_\mu, \mathcal{W}_\sigma, \mathcal{B}_\mu, \mathcal{B}_\sigma \in \mathbb{R}^{m \times k \cdots \{h \text{ times}\} \times n}$, with $h$ being the length of the hidden layer and $k$ being the length of each hidden layer). We decompose each parameter matrix into three parts, such as $\mathcal{W}_{\text{in}} \in \mathbb{R}^{m \times k}$ representing shared weights on the input side, $\mathcal{W}_h \in \mathbb{R}^{k \times k}$ representing shared weights in the hidden layer and $\mathcal{W}_c \in \mathbb{R}^{k \times n}$ representing final non-shared classification weights. A novel distribution learning network is employed to obtain $\mathcal{W}_\mu$ for different connections for $\mathcal{W}_{\text{in}}$ and $\mathcal{W}_h$. $\mathcal{W}_\mu$ of $\mathcal{W}_{\text{in}}$ exhibits two connections: one on the input side of length $m$ and the other on the output side of length $k$. Each of the $k$ output nodes receives input from $m$ incoming nodes on the input side. Empirical findings indicate that all $k$-dimensional tensors of size $m$ follow the same trajectory in the parameter space. This is attributed to the fact that all outgoing nodes observe the same set of input nodes with varying weight connections. Consequently, any perturbation on the incoming side equally affects all outgoing nodes. Therefore, during backpropagation, all connections need to be updated similarly. To address this, we propose utilizing a smaller-sized network to generate tensors of size $m \times k$, i.e., $m$ tensors, each of size $k$. This network employs a distribution generator capable of learning the distribution of any $k$-dimensional tensor. The distribution learning network comprehensively captures the trajectory of all $k$-dimensional tensors across all sessions in the continual learning framework, enabling it to generate tensors that are optimal across all sessions. As all $m$ vectors of dimension $k$ move identically in the parameter space, the same network can be employed to generate all $m$ vectors consistently. The identical procedure is iteratively applied to the connections $\mathcal{W}_h$. We use two different networks for the aforementioned problem. First, we use a deep classification Network (DCN) for which wights are sampled from $\mathcal{W} \sim \mathcal{Q}(\mathcal{W})$ parametrized by $\theta = \{\mu, \sigma\}$ which is mapping the input sample to the output label $\mathcal{X} \to \mathcal{Y}$. Simultaneously, we learn another function $\mathcal{G}_\psi$ from $\mathbb{R}^k \to \mathbb{R}^k$, called a Parameter Learning Network (PLN) which can map any uniformly selected random sample $(\mathbb{R}^k)$ to the parameter space $(\mathbb{R}^k)$. Here we should note that the PLN can generate the parameter of the DCN in such a way that the size of the PLN is far less than the size of the parameter layer i.e. the overall

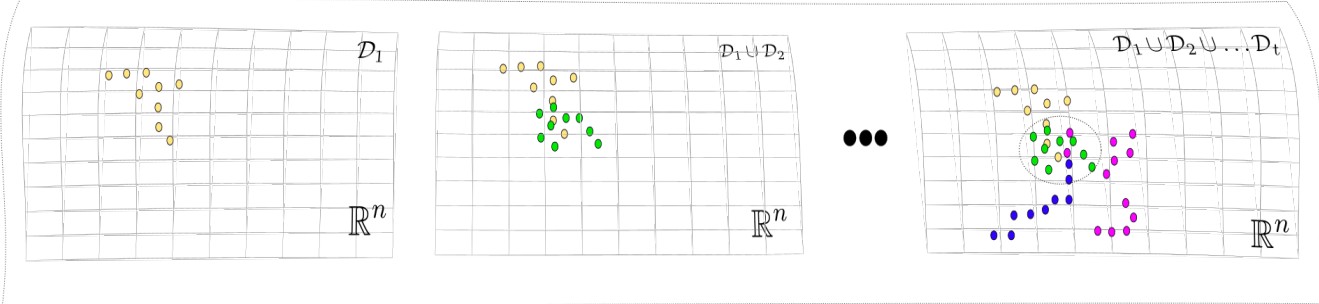

Figure 1: The trajectory of optimal mean parameters within the Bayesian convolutional neural network context involves several key steps. Initially, we derive the trajectories of optimal mean parameters by training the network on dataset $\mathcal{D}_1$, illustrated in yellow. Subsequently, we obtain the optimal mean parameters for dataset $\mathcal{D}_2$, represented in green. Finally, upon training the dataset on $\mathcal{D}_t$, we observe an overlap of optimal parameters within the parameter hyperspace. Consequently, any parameter selected from this space becomes optimal for all sessions, aligning with the overarching objective of continual learning.

parameter size of PLN $\ll \mathbb{R}^k$. Hence, in this way, we can store less number of parameters compared to other Bayesian based techniques as we have to store means and variance of the gaussian distribution in these cases as compared to point-based methods. By composing the two functions we get $f_{\mathcal{W},\theta}(X) = \mathcal{G}_\psi\{\mathcal{Q}(\mathcal{W}/\theta)\}(x)$, which represent our entire model for the Continual Learning Classification tasks.

## 4.2 Efficiently updating parameters of a Convolutional neural network:

Let us consider a Convolutional Neural Network with the input tensor $\mathcal{X} \in \mathbb{R}^{(C_i \times h_i \times w_i)}$ and filters $\mathcal{W}_\mu \in \mathbb{R}^{m \times (l \times k \times k)}$. Here $k$ is the kernel size and $l$ is $C_i$ for the initial convolutional layer and $m$ and $l$ increases as we go deeper into the convoutional layer. After passing it through different convolutional layers it is finally fed to fully connected layers before feeding it to the final application-specific classification head. We try to focus on the drifts of the initial convolutional layers as next fully connected layers can be used similar to what explained in the previous section. For $\mathcal{W}_\mu$ of each convolutional layer we have a different representation matrix. $\mathcal{W}_\mu \in \mathbb{R}^{m \times (l \times k \times k)}$ is reshaped to $\mathcal{W}_\mu \in \mathbb{R}^{m \times lk^2}$ generate $m$ one dimensional tensor to be stored in the representation matrix. The representation matrices for explained in the subsequent section. The parameter drift in the parameter space for the convolutional layers is different than the layers of a fully connected layers. The flattened convolutional masks shows overlapping pattern in the parameter space. Hence in this scenario we focus on the trajectory of the convolutional layers.

**Optimal parameter trajectory in parameter hyperspace:** To ascertain the parameters of $\mathcal{Q}(\mathcal{W}/\theta)$ and store them in the parameter space, the objective is to learn the optimal $\mathcal{G}_\mathcal{W}$ with $\mathcal{W}^{(i)} \sim \mathcal{Q}(\mathcal{W}^{(i)})$, where $\mathcal{W}^{(i)} \in \mathbb{R}^m$. This involves utilizing the trajectories of $\mathcal{S}_k$ for $\mathcal{W}^{(i)}$ across all previous sessions $0, 1, \cdots, t-1$. The focus is specifically on the optimal parameter paths traversed towards the conclusion of training. These paths, located in proximity to the global minima of the loss function, are retained and revisited subsequently to obtain optimal parameter values suitable for all sessions. In this context, we define $\mathcal{T}_{j,k} = \{\mathcal{W}_{j,1}, \mathcal{W}_{j,2}, \cdots, \mathcal{W}_{j,k}\}$ a random trajectory of length $k$ sampled from the parameter space at the $j$-th session when the model is trained on dataset $\mathcal{D}_j$. Finally, $\mathbb{P}(\mathcal{T}_{j,k}|\mathcal{D}_j)$ gives a distribution overall trajectories of length $k$ that can be sampled from the parameter space when the model is trained on the full dataset $\mathcal{D}$. Our objective is the find the trajectory $\mathcal{T}_{\text{opt}}$ which is in minimum distance from all the previously seen trajectories $\mathcal{T}_{1,k}, \mathcal{T}_{2,k}, \cdots, \mathcal{T}_{(i-1),k}$ and the current trajectory $\mathcal{T}_i$ or the overlapping portion of the trajectory $\mathcal{T}_{\text{opt}} = \arg\min_j \bigcap_{i \in \text{tasks}} \|\mathcal{T}_j - \mathcal{T}_{i,k}\|_2$.

This is depicted in Fig. 1. We specifically select the optimal parameters at the conclusion of epochs to fulfill this objective, as illustrated in Algorithm 1.

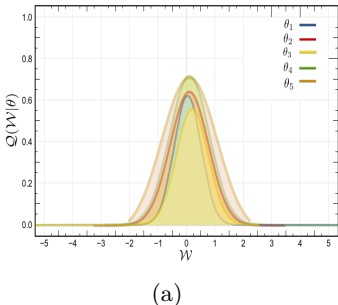
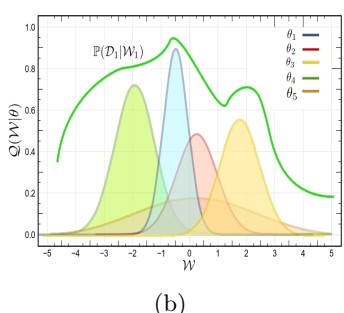
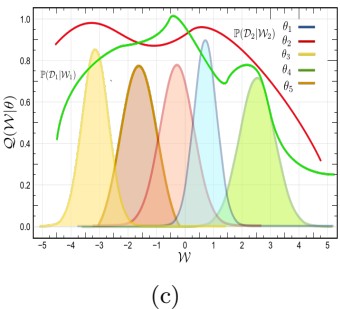

(a)                  (b)                  (c)

Figure 2: Illustration of backward knowledge transfer. (a) parameters at the beginning (b) parameters after training on task 1 (c) parameters after training on task 2. The parameters that exhibited uncertainty after task 1 and were subsequently learned during task 2 contribute to the model's ability to approach a closer alignment between the log evidence curve and its surrogate.

**Representation Matrix Decomposition and Projection:** For a layer $l$, the representation matrix is denoted as $\mathcal{R}_l^t = [\mu_{l,1}^1, \mu_{l,2}^1, \cdots, \mu_{l,p}^1, \cdots, \mu_{l,1}^t, \mu_{l,2}^t, \cdots, \mu_{l,p}^t]$, where $\mu_{l,i}^t \forall i = 1, \cdots, p$ are the samples of layer $l$ at time $t$. Singular Value Decomposition is then applied to the representation matrix, i.e., $\mathcal{R}_l^t = U_l^t \Sigma_l^t (V_l^t)^T$. The determination of the most significant bases relies on the eigenvalues, as defined in Section A.11. Therefore, we opt for the highest singular vectors of the matrix $U$ to acquire the orthonormal basis for the column space of $\mathcal{R}_l^t$. Subsequently, the bases up to the preceding sessions and the current session are concatenated in a matrix $\mathcal{O}_l^t = \mathcal{O}_{\mathcal{C}}^l \bigoplus \mathcal{O}_{\mathcal{D}}^l$. Common bases are particularly valuable as they contribute to optimal performance across all sessions. Thus, common bases are stored in a matrix $\mathcal{O}_{\mathcal{C}}^l$ for common subspace, while the remaining bases are stored in $\mathcal{O}_{\mathcal{D}}^l$ for differentiated subspace. Notably, $\mathcal{O}_{\mathcal{C}}^l \perp \mathcal{O}_{\mathcal{D}}^l$ as the bases in $\mathcal{O}_{\mathcal{D}}^l$ are learned in the absence of correspondence between the network parameter and the dataset, providing novel information about the current dataset. The space complexity is of the order $O(pt) \ll O(lk^2t)$ where $k$ is the number of stored bases, instead of $lk^2$.

$\mathcal{H}_t \rightarrow \mathcal{O}_{\mathcal{D}}^l \bigoplus \mathcal{O}_{\mathcal{C}}^l$ : In this context, let $\mathcal{V}$ represent the space over which the dataset $\mathcal{D}$ is defined. The subspace $\mathcal{H}_l^1$ denotes the subspace over which the parameters $\mathcal{W}_l^1$ corresponding to $\mathcal{D}_1$ are defined. Subsequently, $\mathcal{H}_l^2$ is constructed from $\mathcal{D}_2$ in such a way that $\mathcal{H}_l^2 \perp \mathcal{H}_l^1$, i.e., it has a basis that is not present in $\mathcal{H}_l^1$. Similarly for $\mathcal{H}_l^3$, $\mathcal{H}_l^3 \perp \mathcal{H}_l^1$ and $\mathcal{H}_l^3 \perp \mathcal{H}_l^2$ are constructed from $\mathcal{D}_2, \mathcal{D}_3$ and so on. Now, we can represent $\mathcal{V} = \mathcal{H}_l^1 \bigoplus \mathcal{H}_l^2 \bigoplus \cdots \bigoplus \mathcal{H}_l^t$ since it satisfies the condition outlined in Section A.10. Hence at time step $t$ we have bases $\text{span}(\mathcal{O}_{\mathcal{D}}^l \bigoplus \mathcal{O}_{\mathcal{C}}^l) \subset \text{span}(\mathcal{H}_1 \bigoplus \mathcal{H}_1 \bigoplus \cdots \bigoplus \mathcal{H}_{t-1})$. Moreover, $\mathcal{H}_l^t = \text{span}\{\vec{e}_{\mathcal{C},1}, \vec{e}_{\mathcal{C},2}, \cdots, \vec{e}_{\mathcal{C},n}, \vec{e}_{\mathcal{D},1}, \vec{e}_{\mathcal{D},2}, \cdots, \vec{e}_{\mathcal{D},m}\}$ where $\vec{e}_{\mathcal{C},1}, \vec{e}_{\mathcal{C},2}, \cdots, \vec{e}_{\mathcal{C},n}, \vec{e}_{\mathcal{D},1}, \vec{e}_{\mathcal{D},2}, \cdots, \vec{e}_{\mathcal{D},m}$ are orthonormal bases of $\mathcal{O}_{\mathcal{D}}^l \bigoplus \mathcal{O}_{\mathcal{C}}^l$. Now, $\dim(\mathcal{H}_t) \gg \dim(\mathcal{O}_{\mathcal{D}}^l \bigoplus \mathcal{O}_{\mathcal{C}}^l)$, and we are projecting $\mathcal{H}_t$ onto $\mathcal{O}_{\mathcal{D}}^l \bigoplus \mathcal{O}_{\mathcal{C}}^l$, extending $\mathcal{O}_{\mathcal{D}}^l$ with the bases of the error vectors. We take a Linear Transformation matrix $\Phi : \mathcal{H}_t \rightarrow \mathcal{O}_{\mathcal{D}}^l \bigoplus \mathcal{O}_{\mathcal{C}}^l$. Now, if a transformation is linear, then the corresponding coordinates are also linear maps. Let $\vec{e}_1, \cdots, \vec{e}_n$ be the basis vectors of $\mathcal{H}_t$. Next, we obtain the projection vectors of the basis set of $\mathcal{H}_t$ onto the subspace and $\mathcal{O}_{\mathcal{D}}^l \bigoplus \mathcal{O}_{\mathcal{C}}^l$. Those projection vectors are used to form the $\mathcal{A}_l^t$, i.e, $\mathcal{A}_l^t = [\Phi(\vec{e}_1) \cdots \Phi(\vec{e}_n)]$ where the map $\Phi(\mathcal{V}) = \mathcal{A}_l^t \mathcal{V}$ where $\mathcal{V} \in \mathcal{H}_t$. Now the co-ordinates are related as $[\Phi(\mathcal{V})]_{\mathcal{B}_{\mathcal{O}_{\mathcal{D}}^l \bigoplus \mathcal{O}_{\mathcal{C}}^l}} = \mathcal{A}_l^t [\mathcal{V}]_{\mathcal{B}_{\mathcal{H}_t}}$ . The projection matrix $\mathcal{P}_l^t$ can be obtained by multiplying the orthogonal matrix with its transpose, i.e $\mathcal{P}_l^t = \mathcal{A}_l^t (\mathcal{A}_l^t)^T$.

**Extended Distribution and log-evidence utilizing Importance Weighting:** In our methodology, we augment the prior distribution from the preceding session through the utilization of importance sampling within the subspaces defined by the common subspace and differentiated subspace, as detailed in Domke & Sheldon (2018) and Sobolev & Vetrov (2019). This augmentation facilitates the derivation of a proposal distribution utilizing the bases of $\mathcal{O}_{\mathcal{C}}^l$ and $\mathcal{O}_{\mathcal{D}}^l$. By doing so, we achieve a tighter lower bound for Jensen's inequality, as the resulting distribution is more centered towards the mean. The computation of the importance-weighted prior distribution is expressed by the following formula:

$$\mathcal{Q}(\mathcal{W}_{1:\text{M}}^t | \mathcal{D}_{t-1}) = \frac{\mathcal{Q}(\mathcal{W}_{1:\text{M}}^{t-1} | \mathcal{D}_{t-1})}{\frac{1}{\text{M}} \sum_{i=1}^{\text{M}} \frac{\mathcal{Q}(\mathcal{W}_i^{t-1}, \mathcal{D}_{t-1})}{\mathcal{Q}(\mathcal{W}_i^t)}} \tag{4}$$

This generation involves drawing dummy samples from a random vector generated using the subspaces of common subspace and differentiated subspace. This methodology allows us to establish the correlation between the current dataset and the previous weight distribution, aiding in the determination of either $\mathcal{O}_\mathcal{C}^l$ or $\mathcal{O}_\mathcal{D}^l$. The random weight vector $\hat{\mathcal{W}}^t$ can be represented as a linear combination of bases from $\mathcal{O}_\mathcal{C}^l$ and $\mathcal{O}_\mathcal{D}^l$: $\hat{\mathcal{W}}^t = \sum_{i \in \mathcal{O}_\mathcal{C}^l} c_i v_i^{(\mathcal{C})} + \sum_{j \in \mathcal{O}_\mathcal{D}^l} d_j v_j^{(\mathcal{D})}$. Here, $c_i$ represents the coefficients for bases in $\mathcal{O}_\mathcal{C}^l$, $v_i^{(\mathcal{C})}$ denotes the bases from $\mathcal{O}_\mathcal{C}^l$, $d_j$ stands for the coefficients for bases in $\mathcal{O}_\mathcal{D}^l$, $v_j^{(\mathcal{D})}$ signifies the bases from $\mathcal{O}_\mathcal{D}^l$. To quantify the log-likelihood $\mathbb{E}_{\mathcal{Q}(\mathcal{W})}[\log \mathbb{P}(\mathcal{D}_t|\hat{\mathcal{W}}^t)]$ using the above approximation:

$$\mathcal{L}_{\text{COR}}(\mathcal{W}^t, \mathcal{D}_{t-1}) = \mathbb{E}_{\mathcal{Q}(\hat{\mathcal{W}}_{1:\text{M}})}[\log \mathcal{R}_{t,\text{M}}] + \text{KL}[\mathcal{Q}(\mathcal{W}_1^t)||\mathcal{Q}(\mathcal{W}_1^{t-1}|\mathcal{D}_{t-1})] + \text{KL}[\mathcal{Q}(\mathcal{W}_{2:\text{M}}^t|\mathcal{W}_1^t)||\mathcal{Q}(\mathcal{W}_{2:\text{M}}^t)] \quad (5)$$

This helps us to approximate the log-evidence loss on the current dataset for determining the correlation of the dataset and the network parameters. The derivation is provided in the Section A.6. $\mathcal{R}_{t,\text{M}} = \frac{1}{\text{M}}\sum_{i=1}^{\text{M}} \frac{\mathcal{Q}(\mathcal{W}_i^{t-1}, \mathcal{D}_{t-1})}{\mathcal{Q}(\mathcal{W}_i^t)}$ is importance weighted approximation of the random variable $\mathcal{R}_t$.

**Gradient Analysis and Regularization Proposals:** Next, we analyze the individual terms of the gradient.

$$\nabla_{\theta_t}\mathcal{L}(\theta_t, \mathcal{D}_t) = \underbrace{\nabla_{\theta_t}\mathbb{E}_{\mathcal{Q}(\mathcal{W}_t|\theta_t)}[-\log \mathbb{P}(\mathcal{D}_t|\mathcal{W}_t)]}_{\text{A}} + \underbrace{\nabla_{\theta_t}\text{KL}(\mathcal{Q}(\mathcal{W}_t|\theta_t)||\mathcal{Q}(\mathcal{W}_{t-1}|\theta_{t-1}))}_{\text{B}} \quad (6)$$

**Gradient Analysis of the log-evidence:** The gradient of the log-evidence term comes out to be

$$\nabla_\theta \mathbb{E}_{\mathcal{Q}(\mathcal{W}_t|\theta_t)}[-\log \mathbb{P}(\mathcal{D}_t|\mathcal{W}_t)] = \mathbb{E}_{\mathcal{Q}(\mathcal{W}_t|\theta_t)}[-\frac{\mathcal{W}_t - \mu_t}{\sigma_t} \cdot \frac{\log \mathbb{P}(\mathcal{D}_t|\mathcal{W}_t)}{\mathcal{Q}(\mathcal{W}_t|\theta_t)}] \quad (7)$$

The expression represents the expected value of a term that measures the discrepancy between two probability distributions: the first distribution is $\mathcal{Q}(\mathcal{W}_t|\theta_t) \sim \mathcal{N}(\mathcal{W}_t|\mu_t, \sigma_t)$ that is getting updated at the current session, and the second distribution is the data log-likelihood $\log \mathbb{P}(\mathcal{D}_t|\mathcal{W}_t)$. The term inside the expectation is a measure of how well the normal distribution approximates the data log-likelihood, with a penalty for larger deviations from the mean of the normal distribution. This term is weighted by the ratio of the log-likelihood and the normal distribution, which can be interpreted as a measure of how much confidence we have in the normal distribution approximation. Overall, the expression is a way to assess the quality of the normal distribution approximation to the data log-likelihood.

**Gradient Analysis of the KL divergence term:** The KL divergence term $\text{KL}(\mathcal{Q}(\mathcal{W}_t|\theta_t)||\mathcal{Q}(\mathcal{W}_{t-1}|\theta_{t-1}))$ is a measure of information loss when posterior $\mathcal{Q}(\mathcal{W}_t|\theta_t)$ is used instead of prior $\mathcal{Q}(\mathcal{W}_{t-1}|\theta_{t-1})$. The gradient with respect to $\theta_t = (\mu_t, \sigma_t)$ is given as:

$$\nabla_{\theta_t}\text{KL}(\mathcal{Q}(\mathcal{W}_t|\theta_t)||\mathcal{Q}(\mathcal{W}_{t-1}|\theta_{t-1})) = \begin{bmatrix} \frac{\partial \text{KL}}{\partial \mu_t} \\ \frac{\partial \text{KL}}{\partial \sigma_t} \end{bmatrix} = \begin{bmatrix} \sigma_t^{-1}(\mu_t - \mu_{t-1}) \\ -\frac{1}{2}\left(\sigma_t^{-1} - \sigma_t^{-1}(\mu_t - \mu_{t-1})(\mu_t - \mu_{t-1})^T \sigma_t^{-1}\right) \end{bmatrix} \quad (8)$$

The gradient of the KL divergence signifies the extent to which the KL divergence responds to variations in the parameters of the posterior distribution at session $t$ when the parameters or the mean and variance of the Gaussian parameters are marginally perturbed.

**Proposed mean regularization:** The gradient signifies that an increase in the value of $\mu_t$ will proportionally increase the KL divergence between $\mathcal{Q}(\mathcal{W}_t|\theta_t)$ and $\mathcal{Q}(\mathcal{W}_{t-1}|\theta_{t-1})$ and vice-versa. The scale factor $\sigma_t^{-1}$ denotes the extent to which changes in $\mu_t$ influence the KL divergence. If $\sigma_t$ is small, indicating low uncertainty in the posterior, variations in $\mu_t$ will exert a more pronounced impact on the KL divergence and vice-versa. In this context, challenges arise when one layer is regarded as certain while the subsequent layer is deemed uncertain. Updating the uncertain node may lead to the loss of information in both incoming and outgoing nodes, potentially resulting in instances of catastrophic forgetting. To mitigate this, additional regularization is introduced, focusing on the magnitude of the mean change. This supplementary regularization offers regularization to nodes regardless of their uncertainty status. Consequently, we introduce two regularization terms: one penalizing the magnitude of the mean difference and the other penalizing the magnitude of the ratio of the mean difference to the previous sigma. These regularization terms are mathematically formulated as presented in Equation 9.

**Proposed variance regularization:** The gradient of the KL divergence with respect to $\sigma_t$ involves second-order terms, specifically $(\mu_t - \mu_{t-1})(\mu_t - \mu_{t-1})^T$, representing the products of mean differences along both dimensions. These terms reflect the covariance or correlation between alterations in $\mu_t$ and $\mu_{t-1}$. The inclusion of these second-order terms implies that the influence of mean differences is contingent on the covariance between changes in $\mu_t$ and $\mu_{t-1}$. Positive covariance, indicating concurrent changes, strengthens the impact on the KL divergence, suggesting that the model may readily adapt to alterations when $\mu_t$ and $\mu_{t-1}$ exhibit positive correlation. The limitations associated with this term include the fact that an increased variance $\sigma_t$ reduces the responsiveness of the KL divergence to alterations in $\mu_t$ and $\mu_{t-1}$. Furthermore, updating the variance as the inverse of the variance may lead to many parameters becoming more certain over time. Conversely, if we intend to preserve some degree of learnability in parameters, a gradual increase in certainty for certain parameters would be desirable. We add two extra regularization terms: one for the sparsity of the previous variance and the other for the sparsity of the ratio of variance change relative to the previous variance. Introducing sparsity regularization on the previous sigma encourages adaptability for future parameter learning in lifelong learning scenarios. The second term restricts significant changes in variance values, aiming to minimize variations between steps. Essentially, adding this term to the loss function encourages stability or restrained changes in variance values between consecutive steps, which can be advantageous in scenarios where we aim to manage the rate of change or avoid substantial fluctuations in model parameter uncertainty. Therefore, the updated loss function with the added mean and variance regularization terms is expressed as:

$$\mathcal{L}(\theta_t, \mathcal{D}_t) = \mathbb{E}[-\log \mathbb{P}(\mathcal{D}_t | \mathcal{W}_t)] + \underbrace{\|\mu_t - \mu_{t-1}\|_2^2 + \|\frac{(\mu_t - \mu_{t-1})^2}{\sigma^2}\|_2^2}_{\text{revised } \mu- \text{ regularization}} + \underbrace{\|\sigma_{t-1}\|_1 + \|\frac{\sigma_t - \sigma_{t-1}}{\sigma_{t-1}}\|_1}_{\text{revised } \sigma- \text{ regularization}} \quad (9)$$

This loss function is employed for the model parameter updates, while Eq. 5 is utilized to quantify the correlation between the model parameters and the present dataset.

### 4.2.1 Knowledge Transfer and Sample Correspondence:

**Robust correspondence with previous samples:** This enhancement facilitates robust backward knowledge transfer, emphasizing a strong correlation between the current weight parameters and their predecessors. Consequently, we leverage the bases of the common subspace, eliminating the need for additional parameters in the current session, as we can efficiently reuse the bases from the preceding session. Mean updates are performed based on a stringent correlation check, employing the following equation:

$$\rho(\mathcal{L}(\text{Proj}_{\mathcal{O}_{\mathcal{C}}^l}(\mathcal{W}_l^t), \mathcal{D}_t), \mathcal{L}_{\text{COR}}(\mathcal{W}_l^t, \mathcal{D}_{t-1})) \geq \epsilon \text{ and } \mu_l^t \leftarrow \mu_l^t - \lambda \nabla_\mu \mathcal{L}(\mu_l^t) - \alpha \nabla_\mu \mathcal{L}(\mathcal{P}_l^t \mu_l^t) \quad (10)$$

In this context, $\epsilon$ functions as the correlation threshold, calculated utilizing Equation 22. In this scenario, updating the loss using the bases of the common subspace is feasible. However, this approach may result in a suboptimal approximation for the current dataset in subsequent iterations, as unique information about the dataset remains undiscovered. To address this limitation, SVD is applied to the residual representation matrix $\bar{\mathcal{R}}_l^t = \mathcal{R}_l^t \mathcal{O}_{\mathcal{C}}^l (\mathcal{O}_{\mathcal{C}}^l)^T$ to extract novel information, and these novel bases are stored in the differentiated subspace. The connection between equations 7 and 22 lies in the fact that the second equation can be seen as a measure of how much the gradients of the current task resemble or differ from those of the previously learned tasks. If the gradients of the current task and the stored tasks are highly correlated, it suggests that the model is reusing previously learned knowledge, which is desirable in continual learning. On the other hand, if the correlation is low, it indicates that the model is struggling to transfer knowledge between tasks and may be suffering from catastrophic forgetting. It is also pertinent to observe the layerwise correlation among the weight parameters. In the preceding discussion, we updated the parameters of a specific layer while maintaining the parameters of other layers fixed. Notably, the layer-wise correlation is independent of the dataset. Formally, it can be expressed as: $\rho(\mathcal{L}(\mathcal{W}_{i,\mathcal{C}}^t, \mathcal{D}_t), \mathcal{L}(\mathcal{W}_{j,\mathcal{C}}^t, \mathcal{D}_t)) = \texttt{const} \; \forall i, j = 1, \cdots, n$.

**No correspondence with previous samples:** This modifies the weights according to the current task and stores distinctive samples for the current session. This indicates new knowledge learned by the weight parameters and the distinctive bases are stored in differentiated subspace. Hence we set,

$$\|\rho[\mathcal{L}(\text{Proj}_{\mathcal{O}_{\mathcal{C}}^l}(\mathcal{W}_l^t), \mathcal{D}_t), \mathcal{L}_{\text{COR}}(\mathcal{W}_l^t, \mathcal{D}_{t-1})]\| \leq \delta \text{ and } \mu_l^t \leftarrow \mu_l^t - \lambda(\nabla_\mu \mathcal{L}(\mu_l^t)) \quad (11)$$

**Backward knowledge transfer:** Backward Knowledge Transfer characterizes the enhancement of problem-solving capabilities for earlier tasks through the assimilation of crucial information acquired while addressing the current task. This observation suggests an enhancement in performance on the preceding dataset when incorporating bases from the differentiated subspace, which encompasses the newly acquired bases from the current session. This is illustrated in Fig. 2.

$$\mathcal{L}(\text{Proj}_{\mathcal{O}^l_{\mathcal{C}} \oplus \mathcal{O}^l_{\mathcal{D}}}(\mathcal{W}^t_l), \mathcal{S}^j_{t-1}) \leq \mathcal{L}(\text{Proj}_{\mathcal{O}^l_{\mathcal{C}}}(\mathcal{W}^t_l), \mathcal{S}^j_{t-1}) \tag{12}$$

It is important to highlight that, in order to facilitate backward knowledge transfer, a distinct coreset is preserved, akin to the approach in VCL (Nguyen et al., 2017). Here, $\mathcal{S}^j_{t-1}$ represents the datapoints from session $j \leq t-1$. Subsequently, the acquired bases in the differentiated subspace are concatenated with the common subspace, as they prove beneficial for at least two session parameters. In this context, the concept of confidence is employed, wherein the update of the parameter is confined within the established confidence interval. Specifically, weight updating is permissible in any direction within this confidence interval. This approach enhances the performance of both the current session and the preceding session while mitigating the risk of catastrophic forgetting.

**Forward knowledge Transfer:** In this scenario, the previously acquired bases in the differentiated subspace play a pivotal role in enhancing performance during the current session. Consequently, the bases from $\mathcal{O}^l_{\mathcal{D}}$ are transferred to $\mathcal{O}^l_{\mathcal{C}}$, given their utility for at least two sessions.

$$\mathcal{L}(\text{Proj}_{\mathcal{O}^l_{\mathcal{C}} \oplus \mathcal{O}^l_{\mathcal{D}}}(\mathcal{W}^t_l), \mathcal{D}_t) \leq \mathcal{L}(\text{Proj}_{\mathcal{O}^l_{\mathcal{C}}}(\mathcal{W}^t_l), \mathcal{D}_t) \tag{13}$$

## 5 Experimental Results:

### 5.1 Experimental setup:

**Datasets:** Our experimental configuration is designed to emulate a class incremental learning paradigm, where each session involves a fixed number of classes from a designated dataset. To enhance our understanding, we extend our investigation to a multi-task learning framework, with each session centered around a distinct dataset. We conduct our experiments on two variations of the MNIST dataset, namely Permuted MNIST and split MNIST. Additionally, we explore three distinct variants of the CIFAR dataset, including CIFAR-100, split CIFAR 10-100, alternating CIFAR 10/100, and an 8-mixture dataset. Notably, no data augmentation techniques are employed in our experimental protocol.

**Baseline:** We compare our results with established Bayesian Continual Learning methodologies, as well as non-Bayesian methods, both necessitating the retention of prior parameters. For non-Bayesian methods, our comparison includes HAT Serra et al. (2018) and Elastic Weight Consolidation (EWC) (Kirkpatrick et al., 2017) (although EWC can be regarded as Bayesian-inspired). Additionally, we compare against state-of-the-art methods such as Synaptic Intelligence (SI) Zenke et al. (2017), Incremental Moment Matching (IMM) Lee et al. (2019), Learning without Forgetting (LWF) Li & Hoiem (2017), Less Forgetting Learning Jung et al. (2016), Progressive Neural Network (PNN) Rusu et al. (2016), and PathNet Fernando et al. (2017) for CNN-based experiments. Within the Bayesian framework, we evaluate Uncertainty-based Continual Learning with Adaptive Regularization (UCL) Ahn et al. (2019), as well as Variational Continual Learning (VCL) Nguyen et al. (2017), and two VCL variants, Gaussian natural gradients (VCL-GNG) Tseran et al. (2018) and Variational Adam (VCL-Vadam) Tseran et al. (2018). Additionally, we compare against the latest work within this framework, namely, Uncertainty-guided Continual Learning with Bayesian Neural Network (UCB) Ebrahimi et al. (2019). In this context, we conduct comparisons with ST-BCNN, ST-BFNN, ST-CNN, ST-FNN, which represent Bayesian CNN (Shridhar et al., 2019), Bayesian FNN, and conventional point-based fully connected neural network and convolutional neural network, respectively. These models are trained on new data without any mechanisms in place to mitigate catastrophic forgetting. Additionally, we compare against the combined training of these variants, denoted as CT-BCNN, CT-BFNN, CT-CNN, CT-FNN. These models are trained on the full dataset in a single-step supervised learning fashion, serving as an upper bound as they do not experience forgetting due to their training setup.

| Method | BWT↑ | ACC↑ |
|---|---|---|
| VCL-GNG | - | 90.50 |
| VCL-Vadam | - | 86.34 |
| VCL | -7.90 | 88.8 |
| IMM | -7.14 | 90.51 |
| EWC | - | 88.2 |
| HAT | 0.03 | 97.3 |
| UCL | - | 94.5 |
| UCB | 0.03 | 97.42 |
| LWF | -31.17 | 65.65 |
| ST-BFNN | -0.58 | 90.01 |
| **EVCL-OPT** | **0.03** | **97.8** |
| CT-BFNN | 0.00 | 98.12 |

(a)

| Method | BWT↑ | ACC↑ |
|---|---|---|
| VCL-GNG | - | 96.5 |
| VCL-Vadam | - | 99.17 |
| VCL | -0.56 | 98.2 |
| IMM | -11.2 | 88.54 |
| EWC | -4.20 | 95.78 |
| HAT | 0.00 | 99.59 |
| UCL | - | 99.7 |
| UCB | 0.00 | 99.63 |
| ST-FNN | -9.18 | 90.6 |
| ST-BFNN | -6.45 | 93.42 |
| **EVCL-OPT** | **0.00** | **99.7** |
| CT-BFNN | 0.00 | 99.88 |

(b)

| Method | BWT↑ | ACC↑ |
|---|---|---|
| LWF | -37.9 | 42.93 |
| PathNet | 0.00 | 28.94 |
| LFL | -24.22 | 47.67 |
| IMM | -12.23 | 69.37 |
| HAT | -0.04 | 78.32 |
| PNN | 0.00 | 70.73 |
| EWC | -1.53 | 72.46 |
| ST-BCNN | -7.43 | 68.89 |
| UCB | -0.72 | 79.44 |
| **EVCL-OPT** | **-0.03** | **80.16** |
| CT-BCNN | 1.52 | 83.93 |

(c)

| Method | BWT↑ | ACC↑ |
|---|---|---|
| LFL | -10.0 | 8.61 |
| PathNet | 0.00 | 20.22 |
| IMM | -38.5 | 43.93 |
| LWF | -54.3 | 28.22 |
| HAT | -0.14 | 81.59 |
| PNN | 0.00 | 76.78 |
| EWC | -18.04 | 50.68 |
| ST-BCNN | -23.1 | 43.09 |
| UCB | -0.84 | 84.04 |
| **EVCL-OPT** | **0.24** | **84.58** |
| CT-BCNN | 0.82 | 84.2 |

(d)

Table 1: average accuracy at the final session of Continual learning on the datasets (a)Permuted MNIST, (b)Split MNIST, (c)alternating CIFAR 10/100, (d)8-mixture dataset

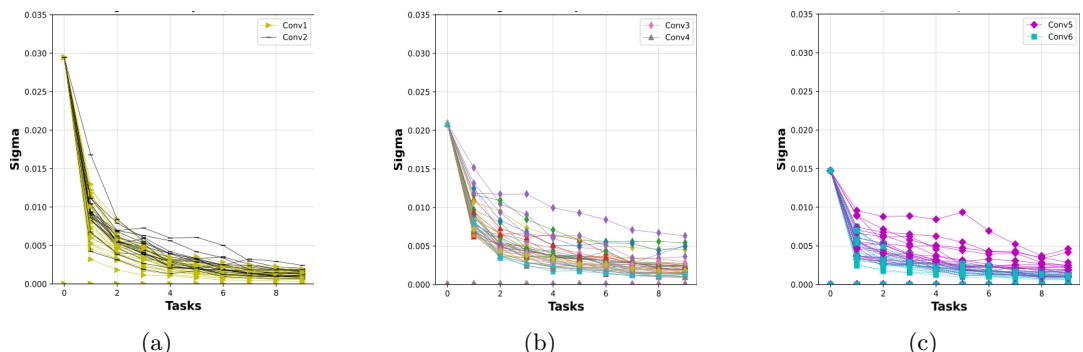

(a)  (b)  (c)

Figure 3: Sigma at different convolutional layers for CIFAR100. (a) layers 1,2 (b) layers 3,4 (c) layers 5,6.

**Performance measurement metrics:** After continual learning, algorithm performance is evaluated using two key metrics: average accuracy (ACC) and backward transfer (BWT), calculated empirically as follows:

$$\text{ACCURACY} = \frac{1}{\text{T}} \sum_{j=1}^{\text{T}} \mathcal{A}_{\text{T},j} \text{ , and } \text{BACKWARD TRANSFER} = \frac{1}{\text{T}-1} \sum_{i=1}^{\text{T}-1} \mathcal{A}_{\text{T},i} - \mathcal{A}_{i,i} \tag{14}$$

Here, $\mathcal{A}_{i,j}$ represents the test accuracy on task $j$ after the model has completed task $i$. The model is trained on all previous tasks after completing the current task. Consequently, we construct a lower triangular matrix $\mathcal{A} \in \mathbb{R}^{\text{T} \times \text{T}}$, where each row corresponds to a timestep, and each column corresponds to a task number. The accuracies for each session are obtained by averaging the rows of the matrix $\mathcal{A}$. BWT quantifies the impact of the learner on previous tasks after being trained on the current task. A positive BWT indicates an improvement in performance on previous tasks, while a negative value suggests catastrophic forgetting. BWT computes the average shift from each diagonal element (representing the accuracy of the model when it initially encounters each task) in the current accuracy matrix.

## 5.2 Main result:

**Permuted MNIST:** Permuted MNIST, a recognized variant of the MNIST dataset commonly employed for assessing continual learning strategies (Nguyen et al., 2017; Kirkpatrick et al., 2017; Zenke et al., 2017), comprises images from the MNIST dataset with randomly permuted pixel values. The dataset $\mathcal{D}_t$ is generated at each time step, incorporating a sequence of 10 random permutations. Table 1a presents the ACC and BWT metrics for EVCL-OPT, benchmarked against state-of-the-art models. After undergoing sequential training across 10 tasks, EWC demonstrate minimal performance distinctions, achieving 90.2%. SI achieves a slightly lower accuracy at 86.0%, as reported by Ahn et al. (2019). HAT attains a slightly better performance (ACC=91.6%), and UCL achieves a decent accuracy of 94.5%. Despite VCL with a coreset showing a

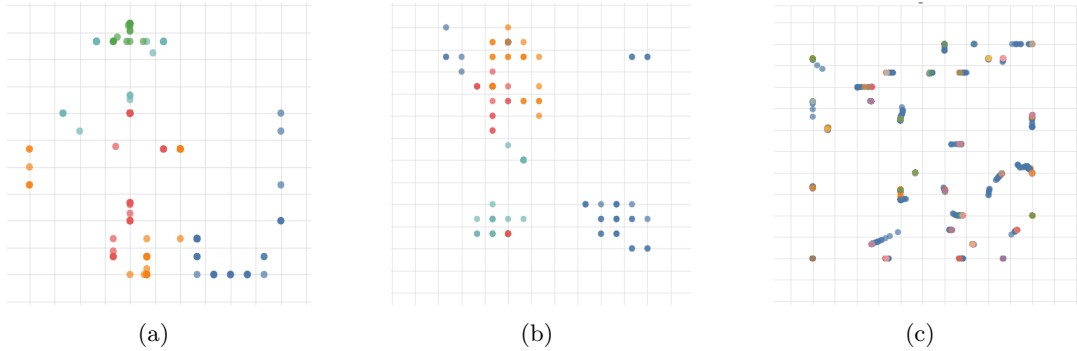

|     (a)     |     (b)     |     (c)     |

Figure 4: Drift of means of weight parameters while training across different sessions at fully connected layers using GTM. (a) Drift of one of $1800 \times 256$ tensor of size 256 each. and (b) Drift of one of $1800 \times 3072$ tensor of size 3072 each. and (c) at convolutional layer. Each color represents a distinct session.

2% improvement, EVCL-OPT outperforms all other baselines with an accuracy of 97.8%. EVCL-OPT outperforms other Bayesian methods in accuracy with minimal forgetting (BWT=0.03%), notably surpassing VCL (ACC=88.80%) with BWT=-7.9%. UCB (ACC=91.44%) initially achieves parity with EVCL-OPT but experiences a decline in performance after task 6, underscoring the constraints of implementing UCB in a single-headed network. When compared to VCL-Vadam (ACC=86.34%), VCL-GNG (ACC=90.50%), and UCL (ACC=94.5%), and EVCL-OPT consistently excels. With 1.9M parameters, Ebrahimi et al. (2019) reports 97.42% for UCB and 97.34% for HAT, while EVCL-OPT achieves 97.8%, close to CT-BFNN (ACC=98.1%), which serves as an upper bound. Importantly, incorporating memory in our experiments for backward knowledge transfer evaluation enhances performance. ST-BFNN, not fully penalized against forgetting, shows reasonable negative BWT, outperforming IMM and LWF.

**split MNIST:** For the split MNIST task, we partitioned the 10 classes into 5 sessions, each comprising 2 classes, following a well-established benchmark for class incremental learning (Nguyen et al., 2017; Kirkpatrick et al., 2017). This benchmark involves a total of 5 tasks, with sequential learning of digits 0-9 in pairs of 2, labeled as 0/1, 2/3, 4/5, 6/7, and 8/9. Table 1b presents results for Bayesian and non-Bayesian neural networks, including simple training (ST-BFNN, ST-FNN) and combined training (CT-BFNN, CT-FNN). Despite the stability of MNIST, Bayesian methods consistently outperform point-based methods, highlighting their effectiveness. EVCL-OPT achieves an average accuracy of 99.7% over the 5 tasks, matching UCL and slightly outperforming UCB and HAT with accuracies of 99.63% and 99.59%, respectively. Significantly surpassing EWC and VCL, EVCL-OPT outperforms all baselines while achieving zero forgetting, akin to HAT. Among VCL variants, VCL-Vadam (ACC=99.17%) outperforms the original VCL (ACC=98.20%) and VCL-GNG (ACC=96.50%).

**Alternating CIFAR 10/100:** We extend the evaluation of EVCL-OPT to three additional CIFAR dataset variants: split CIFAR-100, split CIFAR-10-100 (discussed in the Section A.2), and alternating CIFAR 10/100. In the alternating CIFAR-10/100 setup, CIFAR-10 is distributed across 5 sessions with 2 classes each, while CIFAR-100 is split into 5 sessions, each containing 20 classes. Table 1c displays EVCL-OPT's ACC and BWT outcomes on the alternating CIFAR-10/100 dataset compared to various continual learning baselines. PNN and PathNet exhibit reduced forgetting, albeit at the cost of limited learning in the ongoing session. PathNet's constrained task budget during model initialization impedes learning distinct attributes in each session. Notably, LWF and LFL show limited improvement after training in the initial two sessions on both CIFAR-10 and CIFAR-100, as the initial feature extractor does not exhibit substantial improvement in subsequent sessions. IMM achieves nearly the same performance as ST-BCNN, suggesting that the additional mechanism for reducing forgetting does not provide significant improvement. EWC and PNN perform slightly better than ST-BCNN, while HAT demonstrates noteworthy improvement with minimal forgetting. HAT achieves commendable performance without explicitly remembering past experiences, yet it is surpassed by our EVCL-OPT (ACC=80.16%), which achieves an ACC almost 1.84% higher than HAT. UCB exhibits marginal forgetting but attains an accuracy of 79.4%, with EVCL-OPT surpassing it by a margin of 0.8%.

Thus, we can conclude that for highly intricate and complex computer vision datasets, especially those involving deeper CNN architectures, EVCL-OPT demonstrates commendable performance.

**8-mixture dataset:** The 8-mixture dataset encompasses eight distinct datasets, including MNIST, not-MNIST, CIFAR-10, CIFAR-100, facescrub, Street View House Number, traffic sign, and fashion-MNIST, distributed across eight different sessions. In this composite dataset, the final classification layers for each session exhibit variations, and these diverse configurations are stored and applied during the testing of the corresponding dataset. This design choice stems from our aim to assess the efficacy of the model beyond the classification layer, recognizing that the requirements of the classification layer may vary across different applications. We present the results for continual learning on the 8-mixture dataset in Table 1d. Utilizing the performance of ST-BCNN (43.1%) as a lower bound reference, we observe that EWC and IMM achieve results close to this lower bound. On the other hand, LFL and LWF exhibit a significant drop in accuracy, indicating their challenges in handling difficult datasets with pronounced catastrophic forgetting. PNN, HAT, and UCB are the only baselines that effectively mitigate catastrophic forgetting in this dataset. EVCL-OPT once again outperforms HAT by 3% and UCB by 0.6%.

**Visualization and Analysis of Parameter Drift in Bayesian Neural Networks:** We utilize Generative Topographic Mapping (GTM) Bishop et al. (1998) to examine the DRIFT of means across different layers for Bayesian-CNN and Bayesian-FNN, providing visual insights into the model parameters, as illustrated in Fig. 4. The means $\{\mu_l^t\}_{l=1}^{\mathrm{L}}$ are stored for each layer across all sessions for visualization purposes. Subsequently, we generate plots that showcase the parameters for a specific layer across all sessions, as demonstrated in the Section A.3 in Fig.5 and Fig.6. These plots visually represent the DRIFT in parameter values. In the case of Bayesian-FNN, we observe that parameters remain confined within a compact region for each session. Notably, the means of all nodes exhibit a similar drift pattern at each layer, even when considering the transpose of different layers (Fig. 4). Further analysis indicates that as the number of tasks increases, the drift of the means diminishes, signifying confinement within a smaller region. This consistent drift pattern is empirically demonstrated for all 3072 nodes in the input layer and 256 nodes in the hidden layer. Conversely, in Bayesian-CNN, parameter values corresponding to different convolutional layers show substantial overlap with parameter values from previous sessions. The consequence is that as more sessions are included in training, the optimal parameters tend to choose a previous value with high probability, as depicted in Fig. 6. This emphasizes the importance of distinguishing between the differentiated and common subspaces for effective continual learning.

**Uncertainty Analysis: Standard Deviations Across Convolutional Layers:** To capture the essence of uncertainty, we present Fig. 3, illustrating the standard deviations for different convolutional layers across various sessions for the CIFAR-100 dataset. We initialize the standard deviations as 0.03, 0.02, and 0.015 for layers 1-2, 3-4, and 5-6, respectively. Notably, for layers 1 and 2, the values drop below 0.015 after session 1, while the reduction in other layers is more distributed, ranging between 30% to 60%. Subsequently, some layers exhibit an increase in variance values for certain nodes in the deeper layers. The significant decrease in uncertainty in the initial layer after the first session suggests increased certainty, given the repetitive features encountered at this layer. On the other hand, the fluctuating nature of uncertainty in deeper layers indicates the possibility of having certain nodes with increased uncertainty without compromising the model's ability to learn distinctive attributes. This fluctuation adds extra flexibility by allowing some attributes to be forgotten, making room for new knowledge. Moving to linear layers and the output layer, where precision is crucial for predictions, the decrease in uncertainty is desired to ensure that any alteration does not adversely impact previous predictions.

## 6    Conclusion:

In conclusion, this study introduces Efficient Variational Continual Learning, leveraging the advantages of Variational Inference. While acknowledging the challenge of a large parameter count, we have successfully minimized this issue. Moreover, we propose a modified regularization method that effectively addresses limitations in existing KL divergence regularization, controlling mean and variance dynamics. Our approach not only ensures stability and faster execution but also outperforms current state-of-the-art methods by a considerable margin.

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
