# OpenReview forum: "Efficient Variational Continual Learning with optimal parameter trajectory in parameter hyperspace"
_TMLR — Withdrawn by Authors_

### Review · Reviewer_4H7d · 2024-01-31

**Summary Of Contributions:**

This paper aims to propose a framework for continual variational inference whose goal is to be more memory efficient than current approaches. Unfortunately the paper is marred with clarity issues to the point its methodological contributions cannot be properly assessed.

**Audience:**

No

**Broader Impact Concerns:**

None.

**Claims And Evidence:**

No

**Requested Changes:**

Do a thorough re-write of the entire paper.

**Strengths And Weaknesses:**

**Strengths**

Unfortunately, I found none.

**Weaknesses**

The paper is extremely poorly written, to the point that I do not believe the authors bothered to proofread their work before submitting it, which makes the paper mostly incomprehensible. To give some examples:

- Throughout the paper the authors use notation without properly defining it, page 6 being the worst offender.

- The authors attempt to carry out mathematical explanations in plain language with no accompanying equations, e.g. sec 4.1.

- Figure 1, whose goal is to illustrate the method is also extremely poorly explained.


Besides this fundamental lack of clarity, the the paper is full of typos/errors which also make reading the paper harder, e.g.:

- The authors mix up "|" and "/" when for the conditioning symbol in conditional probability.

- The authors reference an algorithm, figures, and sections which are, at a first glance, nowhere to be found. Upon closer inspection, the authors submitted the appendix as supplementary material in a zip file, rather than after the references as per TMLR's formatting guidelines.

- The authors use \citet{} and \citep{} interchangeably in LaTeX.

---

### Review · Reviewer_yDqD · 2024-02-07

**Summary Of Contributions:**

Unfortunately, I could not fully understand the paper. The setup resembles Variational Continual Learning [Nguyen et al, 2017] in that approximate posteriors are transferred across tasks. The authors propose some networks (similar to hyper-networks as far as I can understand) for updating the parameters of a fully connected network. For convolutional nets, they rely on parameter trajectories obtained during continual learning. This is followed by an SVD-based compression of parameters from different tasks. Finally, an importance sampling based approximation for KL divergence is introduced.

**Audience:**

Yes

**Broader Impact Concerns:**

-

**Claims And Evidence:**

No

**Requested Changes:**

### Writing
The writing should be improved:
  - to increase clarity (for example, contributions are not clear in the abstract)
  - citations and parenthesis should be checked (for example, see the second paragraph: _"... Forgetting Kirkpatrick et al 2017,..."_ ---> _"... Forgetting [Kirkpatrick et al 2017,...]_"
  - more recent approaches, e.g. based on pre-trained networks, should be discussed in the intro.
  - What does $\theta$ stand for in (1)? It's used to denote both prior and posterior parameters.
  - What does / stand for at the end of page 3 and rest of the paper?
  - What is $\alpha$ above (3)?
- It is not clear how the authors arrive at (3). I believe the standard normal prior in VI is replaced with the variational posterior of the previous task but it's not clear and lacks citation to "Variational Continual Learning" paper. Likewise, (3) has the expected likelihood term while (1) and (2) don't, this should also be written explicitly.
- Notation is unnecessarily re-introduced in section 4.1.
- _times_ written in words in section 4.1.
- _h being the length of the hidden layer and k being the length of each hidden layer)_ is not clear.

### Minor
- This does not hold for all Bayesian inference. The context should be made clear: _Bayesian techniques handle uncertainty in parameters by representing each parameter as samples drawn from a distribution defined by its mean and variance._
- VI typically assumes a fixed prior: _Variational Inference (Hoffman et al., 2013) is the most famous, which approximates the prior and posterior distribution by minimizing the distance between the true and the approximate distribution..._
- _the weight is drawn_ ---> _the weights are drawn_
- _Baye’s rule_ ---> _Bayes rule_
- In (2), no need for the first expectation as the marginal is independent of the variational approximation.

### Questions
- What does "problem-solving abilities" mean at the end of the first page? Does this question differ from _forward transfer_?
- What does _non-shared classification weights_ mean? What modules share the weights?

**Strengths And Weaknesses:**

### Weaknesses
- The approach is not explained clearly. What are the optimization objectives? Do they change after every task? Does the work only apply to CNNs and only 1-layer MLPs? Do different circles in Fig 1 belong to different runs? What is $S_k$ on page 5? Is the session number denoted by a subscript $W_{j,i}$ or superscript $W^{(i)}$? Does the distance in the parameter space directly translate into function space, given that neural networks are highly non-linear?? How are the paths located in proximity to the global minima, given that the data arrives sequentially (making global optima difficult to reach)?
- The writing and notation must be improved (see requested changes).
- I am not sure if parameter storage is a huge bottleneck. I agree that variational approximation requires twice the number of parameters but it doesn't explode with the number of tasks unlike, e.g., growing architecture based methods. So I don't see this becoming a problem in the limit.

---

### Review · Reviewer_Pqwy · 2024-02-26

**Summary Of Contributions:**

The paper proposes a novel Bayesian neural network (BNN) learning method for class-incremental learning. The paper is a broad mix of many different ideas, which for the most part are explained in a very confused way (this includes hyper-networks, regularization terms, matrix decompositions, bases concatenation, etc.). Hence, it is really difficult to understand the contribution of the paper. The experiments seem to validate the proposal, but they focus only on accuracy, disregarding aspects of memory, parameter's count, time, etc.

**Audience:**

No

**Broader Impact Concerns:**

N/A.

**Claims And Evidence:**

No

**Requested Changes:**

Based on my previous comments, I think the paper should undergo a complete rewriting of its content. I also suggest ample proofreading and double-checking for clarity and exposition. There might be valuable content, but at the moment it is impossible to evaluate it objectively.

**Strengths And Weaknesses:**

I have found the paper very difficult to follow and understand. I will list some of my concerns as they appear in the paper, but I stress that, without a full rewriting, this paper is unacceptable for publication in my opinion.

- Abstract / introduction: they are both very broad, with vague sentences like "*We present an augmented Evidence Lower Bound term*", or "*an efficient method for storing fully connected layer parameters*", but with no details. As such, reading the introduction is basically useless. On top of that, citations are used incorrectly (citet / citep).

- There are many terms used without an explanation, and sentences that lead nowhere. Take this sentence: "*Subsequently, through meticulous experimentation, we visually illustrate the progression of the posterior distribution converging towards the optimal parameter distribution that is universally effective across all sessions.*" First, what is a "session"? I assume this corresponds to a "task" in the CL literature. Also, where is the "meticulous experimentation" in the paper? The closest I could find is a very generic depiction in Fig. 2.

- Organisation is also weird. For example, in the introduction: "*Variational inference stands out as a prominent optimization-based method, enjoying widespread popularity in this category*"; but the entire paragraph before was discussing variational inference. Similarly later on: "*Consider a fully connected neural network operating in a supervised learning scenario*", but the scenario was already introduced in the section before.

- Related works are discussed but their relationship with this paper is completely ignored. Some are also dismissed in a very harsh way, e.g., "*In these methods, storing previous network parameters is essential, but none address efficient storage to reduce costs or establish universally beneficial knowledge across sessions.*" No method in CL has ever addressed efficient storage? There are even papers on subspace decompositions - that the authors are citing - falling in this category.

- Math notation is strange and there are many errors. For example, let me highlight some points from Section 3 / 3.1 (the only one which has a series of connected equations):

1. P3: the prior is denoted as $\mathbb{P}(\mathcal{W} \mid \theta)$, but $\theta$ are used as the variational parameters (for $\mathbb{Q}$), which is confusing.
2. In (2), the term inside the expected value ($\mathbb{P}(\mathcal{D})$) should be $\mathbb{P}(\mathcal{D} \mid \mathcal{W})$, otherwise how can we take an expected value with respect to the variational distribution? This is also inconsistent with the same equation in (3) (which is instead correct).
3. In the same equation the prior becomes $\mathbb{P}(\mathcal{W} \mid \mathcal{D})$ (inside the KL divergence) which is also confusing, because how can the prior be conditioned on the dataset?
3. At the end of the page "and $ \mathbb{P}(\mathcal{W}_{t-1} \;\lvert\; \theta, \tilde{\mathcal{D}}) $" but this term was not used in the equation before.

- Many similar comments can be made in the rest of the paper, for example, what is $\mathcal{B}$ in Section 4.1? Or "*utilizing the trajectories of $\mathcal{S}_k$", what is $\mathcal{S}_k$*?

- Moving on to the main technical content of the paper. In Section 4.1, for fully-connected networks they propose to model the variational terms via a separate "*Parameter Learning Network (PLN)*", which "*takes a vector of size $k$ and outputs a vector of size $k$*". If I understand correctly, the input is supposed to be the output of the previous layer, and the output the weights for a single neuron? This network is then sampled "m" times (the number of neurons), under the assumption that "*all k-dimensional tensors of size m follow the same trajectory in the parameter space*", given "*empirical evidence*". This is a huge assumption and this "empirical evidence" is never given. It does not help that math has now disappeared and everything is discussed with textual descriptions which are hard to parse.

- How is this PLN parameterized? This is never discussed in the paper.

- Sec. 4.2, for a CNN the authors use a different approach which is based on "*weight trajectories*". I do not understand if they are referring to gradient descent trajectories? Markov chain Monte Carlo sampling? Or other things? Fig. 1 is useless (and also, why is it curved?). Algorithm 1 is not in the main text, it's at the end of the appendix but it is so long (> 1 page) to be useless. Also, if you should store $k$ times the weights, don't you need an enormous amount of memory?

- The basis decomposition is so unclear and vague that I have very little idea of what is happening. However, the network's size seems to increase with the size of the tasks? This is also not discussed.

- The entire "*Gradient Analysis of the KL divergence term*" is a general description of the ELBO term. Take this sentence as an example: "*The gradient of the KL divergence signifies the extent to which the KL divergence responds to variations in the parameters of the posterior distribution at session t when the parameters or the mean and variance of the Gaussian parameters are marginally perturbed.*" It has the feel of being written by an LLM.

- The rest is also completely unreadable. For example, take (10). Half of the terms are not explained (what is $\rho$? What is $\mathcal{L}_{\text{COR}}$)? It seems like the gradient updates are restricted based on some projections but this is really hard to parse.

- In all these sections a lot of useful content has been randomly moved to the Appendix, e.g., "*since it satisfies the condition outlined in Section A.10*", but A.10 is just a single equation.

- "*in order to facilitate backward knowledge transfer, a distinct coreset is preserved*": does this mean you store previous data? This was explicitly negated in the introduction ("*In our approach, we do not include the data from past sessions for training the model*").

These are just glaring points, many more imprecisions and unclear content can be found in the paper.

---

### Note · Authors · 2024-03-10

I have read and agree with the venue's withdrawal policy on behalf of myself and my co-authors.